# Tribal Tobacco Use Project II: Planning, Implementation, and Dissemination Using Culturally Relevant Data Collection among American Indian Communities

**DOI:** 10.3390/ijerph19137708

**Published:** 2022-06-23

**Authors:** Kendra M. Roland, Madison D. Anderson, Dana M. Carroll, Anna G. Webber, Kristine L. Rhodes, John Poupart, Jean L. Forster, Melanie Peterson-Hickey, Wyatt J. Pickner

**Affiliations:** 1American Indian Cancer Foundation, Minneapolis, MN 55113, USA; kendraroland21@gmail.com (K.M.R.); manderson@americanindiancancer.org (M.D.A.); agwebber@gmail.com (A.G.W.); 2Division of Epidemiology and Community Health, School of Public Health, University of Minnesota, Minneapolis, MN 55454, USA; forst001@umn.edu; 3Division of Environmental Health Services, School of Public Health, University of Minnesota, Minneapolis, MN 55455, USA; dcarroll@umn.edu; 4Asemaake, St. Paul, MN 55104, USA; kris@asemaake.com; 5American Indian Policy Center, President (Ret.), Phoenix, AZ 85004, USA; ogema2001@yahoo.com; 6Minnesota Department of Health, St. Paul, MN 55164, USA; melanie.peterson-hickey@state.mn.us

**Keywords:** community-based participatory research, culturally tailored data collection, tobacco use in American Indian communities, surveillance

## Abstract

American Indians have substantially higher commercial tobacco-related cancer rates when compared to the general population. To effectively combat commercial tobacco-related cancer, it is important that tribal nations obtain current and accurate community-specific data on commercial tobacco use and exposure-related attitudes and behaviors. With the goal to collect, synthesize, and disseminate data on tobacco use, including the role traditional tobacco plays among American Indian people, the American Indian Cancer Foundation (AICAF) and various stakeholders developed and implemented the Tribal Tobacco Use Project II (TTUP II) during 2018–2021. Building upon its predecessor, the Tribal Tobacco Use Project I (TTUP I), TTUP II used principles of community-based participatory research and culturally appropriate methods, such as Reality-Based Research, in partnership with tribal nations. We describe the TTUP II rationale, methods for participant recruitment and data collection, emphasizing the importance of using culturally relevant survey items to disentangle commercial tobacco use from traditional tobacco use. American Indian traditional tobacco is viewed as medicine in these communities with a unique socio-cultural context that must be addressed when engaging in commercial tobacco control efforts in American Indian communities. This approach may be useful to other tribal nations who are interested in conducting culturally relevant tobacco surveillance efforts.

## 1. Forward

TTUP II cultivated good partnerships with its methods and applications of modern research with American Indian communities. Modern research is extremely difficult to “retrofit” into traditional American Indian culture. One issue in particular is that American Indians view tobacco as “sacred” and it is not to be used casually. How does one research such a sacred belief while maintaining respect for the culture? This becomes most evident when comparing sacred tobacco and commercial tobacco. We cannot be disrespectful to the sacredness of American Indian tobacco. If it is treated as an “object”, it could do great harm not only to the TTUP II project, but to future projects that involve American Indian communities. To interpret American Indian tobacco in a way that is not respectful to its sacred use and ancient origin would be a mistake.

John Poupart, MPA, Anishinaabe, Community Oversight Group Member.

## 2. Introduction

*Asemaa. Cansasa. Sacred/Traditional Tobacco.* These words may look unfamiliar to many, but to the Anishinaabe/Ojibwe and Dakota people, whose tribal lands lie within the boundaries of the state of Minnesota, these are words of ancestral origin that hold spiritual and sacred significance. These words translate to “tobacco” in various languages and dialects and refer to tobacco in its purest form, as a plant, *Nicotiana rustica* (tobacco), and *Cornus sericea* (red osier dogwood) [1,2]. American Indian people have a unique relationship with tobacco, making it important to discuss the traditional and sacred role that tobacco plays in their lives. Through oral tradition and storytelling, American Indian people understand that their ancestors had a healthy relationship with tobacco, using it as a medicine for spiritual, emotional, mental, and physical guidance and healing [3]. Traditional tobacco has many uses: to heal, bless, or smudge, as a gift of welcoming, for prayer, for purification, and for cleansing in use with other plants, such as sage, cedar, and sweet grass. Dependent upon heritage, family teachings, and tribal customs, traditional tobacco takes many different forms. Customs, use, distribution, and the cultivation of such tobacco varies from tribe to tribe.

American Indian people have preserved the values of traditional tobacco while suffering from colonization efforts and religious persecution for hundreds of years [3]. The freedom of religion promised to all in the First Amendment of the United States (U.S.) Constitution was not extended to the land’s original inhabitants [3]. From the late 1880s, until the passing of the original American Indian Religious Freedoms Act of 1978, American Indian people were imprisoned for participating in traditional and spiritual ceremonies [3]. As a form of cultural preservation, American Indian people began using commercial tobacco products in many traditional settings. This substitution has had lasting effects on the relationship of American Indian cultures with both traditional and commercial tobacco use and subsequently on their health [3].

A multitude of reasons can be attributed to the high prevalence of commercial tobacco use among American Indian people. These include but are not limited to: the prohibition of spiritual practices; the tobacco industry’s targeting of American Indians in marketing efforts; misappropriation of American Indian culture and traditional tobacco in tobacco industry marketing campaigns; community/family economic stress; and discrimination and historical traumas, including community massacres, forced relocation, and removal of American Indian children into boarding schools in attempts to assimilate them into Western beliefs, language, and cultural practices. Additionally, while American Indian people report wanting to quit smoking at a similar level as other racial/ethnic groups, quit rates are lower among American Indian people relative to other racial groups [4]. Lower quit success has been attributed to lack of access to cessation services and programs that are tailored to fit the needs and culture of American Indian people who smoke [5]. For all of these reasons, American Indian people experience the highest prevalence of cigarette smoking compared to all other racial/ethnic groups in the United States [6]. For example, based on data from the 2019 National Health Interview Survey, the prevalence of cigarette smoking in 2018 among American Indian/Alaska Native, non-Hispanic people was 20.9% compared to 15.5% among White, non-Hispanic people [6]. For multiple reasons described elsewhere [7], the detrimental effects of commercial tobacco use among American Indians are particularly evident in the Northern Plains region of the United States, including Minnesota, where an estimated 59% of American Indians smoke every day or some days in 2013 relative to 16% of the state’s total population [8]. Consequently, secondhand smoke exposure, which is also linked to numerous negative health outcomes, is high among American Indian people [9]. As a result, American Indians in Minnesota experience rates of tobacco-related morbidity and mortality, such as lung cancer and heart disease, as high as two-fold more than that of White persons [10,11,12].

To be able to effectively combat commercial tobacco use and reduce cancer rates among American Indian people, it is crucial that tribes and American Indian communities obtain current and accurate community-specific data on commercial tobacco use and exposure (e.g., cigarettes, e-cigarettes, secondhand smoke, and smoke-free policy), related attitudes and behaviors, and the role traditional tobacco plays among American Indian people. With the goal to collect, synthesize, and disseminate data on tobacco use in partnership with tribal nations, the American Indian Cancer Foundation (AICAF) and various stakeholders developed the Tribal Tobacco Use Project II (TTUP). This project built upon its predecessor, the Tribal Tobacco Use Project I (TTUP I) [9]. TTUP I was conducted during 2010–2012 through the American Indian Community Tobacco Project’s partnerships with tribal nations and urban communities in Minnesota [9]. TTUP I data generated the above-mentioned 59% prevalence estimate of smoking among Minnesota American Indians. Additionally, TTUP I showed that nearly three-fourths (71%) of Minnesota American Indians had ever used tobacco for ceremonial prayer or in a sacred way [9]—data that have been used by tribes (e.g., “Two Tobacco Ways” Principle Guiding Work in Tribal Nations [13]) and public health organizations (e.g., Clearway Minnesota’s “Keep Tobacco Sacred” campaign) to make the case that tobacco control, which includes surveillance efforts, cannot be conducted without acknowledging the sacred role of tobacco in American Indian culture.

Since the TTUP methods may be useful to other tribal nations and public health organizations who are interested in conducting culturally relevant tobacco surveillance, the objectives of this paper are to (1) highlight the strengths and uniqueness that TTUP provides as an American Indian tobacco surveillance system, and (2) provide an overview of the specific planning, implementation, and dissemination efforts.

## 3. Tribal Tobacco-Use Project—A Culturally Relevant Tobacco Surveillance Effort for American Indian People Developed by American Indian People

TTUP has several unique elements that enhance its relevance. All project activities emphasized the principles of community-based participatory research (CBPR) and reality-based research (RBR) [14,15]. CBPR acknowledges the different ways of knowing and gives equal weight to scientific expressions of knowledge and traditional or cultural expressions of knowledge. CBPR involves community stakeholders in every aspect of research, including identifying key topics or issues; defining terms, strategies, outcomes, and goals; designing data collection instruments; analyzing data and other information; developing strategies and activities for returning the information to the community; and evaluating the outcome of those activities. RBR can be considered an extension of CBPR that is specific to the American Indian community whereby the research respects and incorporates American Indian traditions and understands that American Indian people have a different worldview than other groups. Below, we describe some of these elements in detail.

### 3.1. TTUP Respects and Honors Tribal Sovereignty by Engaging American Indian Tribes and Urban Communities through a CBPR Framework

Federally recognized tribes are sovereign nations; this distinction makes them responsible for the health and wellbeing of the population they serve (i.e., tribal members). As part of their legal right to self-govern, tribes can choose whether to enter into research agreements and determine how the research is conducted to protect against the predatory and harmful use of data. These responsibilities are critical for ensuring the viability and future health of the tribe.

There are instances throughout history where tribal nations have been exploited and suffered various forms of abuse at the hands of researchers [16]. It is the culmination of such experiences that has contributed to justifiable mistrust by some American Indian persons of the scientific community and therefore reluctance to participate in research.

Including the community as a partner in all decisions made for the research [14,17] is requested, if not mandated, by tribes when considering whether they will engage in the research [18,19]. To facilitate engagement and participation from the tribes, study staff engaged in targeted outreach to tribal health directors and various urban American Indian community agencies in Minnesota. Outreach involved numerous meetings and gatherings to build trust and to develop shared ownership of the project goals. It is important to note that commercial tobacco prevention and control was already viewed as a major public health priority to tribal health directors; thus the need for surveillance data was viewed as a community-identified need. Support for TTUP from tribal health directors led to buy-in from tribal leadership and relevant decision makers. As described in the Methods Section, the CBPR process is further embodied through obtaining thorough feedback and approval from each of the participating tribes in all TTUP project components.

### 3.2. Collection of Tribal Affiliation Is a Unique and Critical Aspect of TTUP

Many health surveys consider American Indian people to be a homogenous group and therefore ignore the distinct cultural, social, and historical contributors to health that vary across tribes. While there are reasons to use aggregate data (e.g., when contributors to health are similar across tribes and to enhance sample size), tribal-specific data are more meaningful to individual tribes as they can be used to understand the extent of commercial tobacco use among their own tribal members and to inform their own public health efforts, policies, and resource allocation. Therefore, TTUP collected tribal affiliation from each participant, allowing for tribal-specific data reports presented to tribal leadership. In addition to generating tribally specific data, each participating community ultimately owns their data and can use them as they see fit.

### 3.3. In Contrast to Mainstream Tobacco Surveillance Efforts, TTUP Distinguishes Traditional from Commercial Tobacco Use and Collects Data on Both

Given the history and context of tobacco within the U.S., coupled with the centuries of religious and spiritual persecution, accurately assessing tobacco use among American Indian communities requires capturing the nuances that are often hidden within national surveillance efforts. TTUP uses intentional language, such as “not including ceremonial or sacred use….”, before each question on the topic of commercial tobacco use. Using this type of language ensures accurate capturing of the non-sacred use of tobacco. Additionally, the use of traditional tobacco is assessed in a separate section titled “Ceremonial or Sacred Use of Tobacco”. Participants were asked questions on the type of product used (traditional tobacco or commercial tobacco, including cigarettes, pouch, or loose tobacco), and method of use (burn it outdoors, smoke it, etc.) when using tobacco for sacred or ceremonial purposes. Participants were also asked about the accessibility of traditional tobacco in their communities and if they were interested in learning more about ceremonial or sacred use of traditional tobacco.

### 3.4. TTUP Responds to the Requests and Priorities of the Community

In alignment with the CBPR approach, TTUP was adapted to meet the needs of each unique participating tribe. Each tribe was able to review and make changes to the TTUP survey instrument on an as-needed basis. Flexibility in the mode of data collection (e.g., in-person, virtually) was also paramount to tribal support. TTUP II data collection coincided with the COVID-19 pandemic. Refer to the Methods Section for COVID-19 adaptability protocols.

## 4. Methods: TTUP II Planning, Implementation, and Dissemination

TTUP II was designed to be executed through three phases (planning, implementation, and dissemination) over a three-year period. Implementation phases for participating tribal nations occurred on a rolling basis from October 2017 until July 2021. Each of the phases are described in greater detail below and summarized in Appendix A.

### 4.1. Planning Phase (October 2017 to June 2018)

A nine-month planning period was set aside for the development of project materials and the recruitment of tribal partners. While this timeline may seem long to some, it reflects the emphasis on building trusting relationships, which includes ensuring the tribal partners participate throughout all phases of the planning. The items listed below were developed concurrently.

#### 4.1.1. Community Oversight Group (COG)

A Community Oversight Group (COG) was formed to provide guidance on planning, implementation, and dissemination, with the needs of the tribal partners always at the forefront of decision making. The COG consisted of 11 individuals who represented a diverse group of TTUP II stakeholders. Selection criteria included: having existing relationships and involvement with the public health activities of tribes, academics with relevant expertise, and state department of health tobacco control professionals. The COG met quarterly from the beginning of the project with additional meetings, email, and phone correspondence scheduled as necessary.

#### 4.1.2. Recruitment of Tribal Partners and Urban American Indian Community Stakeholders

Tribal recruitment and relationship building was the main priority of the planning phase. TTUP staff attended a variety of meetings throughout the state for opportunities to present the goals of TTUP II to elected tribal leaders, tribal health directors, and various stakeholders who directly served the tribes in varying capacities. In addition to in-person meetings, email and phone correspondence were a major contributor to securing tribal support and participation. TTUP staff tracked outreach and recruitment efforts with tribal officials and tribal health directors. Recruitment materials, which included TTUP overview presentations that emphasized the potential benefits to the tribe, letters to tribal leadership, TTUP I aggregate findings, and tribal specific findings, if the tribe being recruited had participated in TTUP I, were created and used to recruit and inform tribes about the potential benefits of participating. TTUP staff met with all 11 tribal health directors in the state of Minnesota and held an informational session (in person and/or virtually) to present the TTUP II project and shared recruitment materials. Buy-in from tribal health directors typically led to approval and implementation of the TTUP II study within their respective communities.

In addition to recruiting individual tribes as partners, TTUP also recruited stakeholders who served the urban (Minneapolis/St. Paul Metro) American Indian community in Minnesota where approximately 50% of Minnesota American Indians reside [20]. Tribal agencies that were located within urban zip codes were contacted and relationships were established. These agencies ranged from nonprofits, tribal specific urban offices, or other entities that served the Minnesota urban American Indian population. Through these relationships and correspondence, TTUP staff were able to identify urban American Indian community members to serve as community interviewers for the urban sampling (described below).

#### 4.1.3. Tribal Resolutions and Data Use Agreements

Written research agreements, such as tribal resolutions, can be helpful and are often required by tribes to formalize research collaborations and are a transparent way to describe the mutual goals, responsibilities, and intended research benefits to the various partners. Research agreements should be revisited frequently and updated as needed to ensure accurate reflection of evolving research collaborations [19]. Please refer to the National Indian Health Board (www.nihb.org (accessed on 21 January 2022)) and National Congress of American Indians (www.ncai.org (accessed on 21 January 2022)) for additional considerations. Data use agreements (see Appendix B for template) were also developed to delineate the life cycle of the research data from the type of data being collected, who collects and who has access to the tribal data, how the research data is stored and eventually destroyed, how and to whom the data is disseminated (e.g., publications, reports, presentations), and the approvals required for the dissemination. Only de-identified participant numbers were included with the survey data. Tribal-specific data were to be owned by the participating tribe and it was at their discretion how the tribal-specific data would be used after completion. Tribes were able to opt into being included in a statewide aggregate report to the public. Tribal health directors typically led the efforts in reviewing and providing feedback on the tribal resolution and data use agreement, and securing the appropriate approvals (e.g., through tribal council/government).

#### 4.1.4. Survey Development

The TTUP survey was designed as an interviewer-led tablet-assisted face-to-face survey. The survey instrument collected information on traditional/sacred and commercial tobacco use, knowledge, attitudes, and beliefs of tobacco use and secondhand smoke exposure. Commercial tobacco products included cigarettes, electronic cigarettes, and smokeless tobacco. Several relevant surveys were consulted prior to development including TTUP I, Minnesota Adult Tobacco Survey (MATS), American Indian Adult Tobacco Survey (AIATS), Behavioral Risk Factor Surveillance System (BRFSS), and Adverse Childhood Experiences (ACEs). The TTUP I survey instrument was included in its entirety, maintaining core questions, with the addition of e-cigarette usage, stress, trauma, and resilience of American Indian people. The TTUP II survey was developed with extensive input from the COG. Each tribe had the opportunity to review the survey and add or remove questions based on their discretion. The final instrument included 150 items asked in English language and took approximately 30 min to complete.

#### 4.1.5. Institutional Review Boards (IRBs)

It was critical to ensure that the study instrument and protocol were reviewed by each of the tribal partners for feasibility and cultural appropriateness, as well as to guarantee the privacy of the tribal members who participated and the confidentiality of the tribe at large. In addition to getting statewide IRB approval from the Minnesota Department of Health and the University of Minnesota, we sought review and approval from relevant tribal IRBs (unlisted to protect the identity of participant tribes). After IRB approvals were secured, a data sharing agreement was developed for each participating tribe tailored to their requirements.

### 4.2. Implementation Phase (July 2018 to July 2021)

Multiple steps were performed to facilitate data collection in a standardized way with each participating site.

#### 4.2.1. Hiring and Training of TTUP II Community Specific Personnel

Each participating tribe assigned a site coordinator as the main point of contact throughout the study. Site coordinators were responsible for identifying community members who would be interested in serving as interviewers. For most tribes, individuals already embedded in their public health activities were asked to be the site coordinator and, when applicable, the interviewers. Interviewer training typically lasted around six hours and covered a comprehensive overview of TTUP II and in-depth instruction on the protocol and procedures for the survey, including how to navigate the survey platform (Qualtrics), track participation, and distribute participant incentives. Training sessions were also used to conduct mock interviews. Interviewers were required to sign an interviewer confidentiality agreement and were given study supplies (e.g., incentives, tablets for data entry, pens, paper, etc.). To aid in recruitment, retention, and participation of community members, interviewers and survey participants were compensated for their time and effort. All partners agreed that incentives were appropriate for compensating community members for the knowledge and effort contributed. Thus, for every completed survey, a USD 20 gift card was given to the participant as well as to the interviewer.

#### 4.2.2. Sampling Methods for Tribal Partners

To obtain valid tribal-specific data, a list of all tribal members was required for sampling in each tribal nation. There were two ways that TTUP staff assisted tribes in generating such lists.
Tribal Enrollment Records—each tribal nation has varying protocols and policies set in place for individuals to become enrolled members of their tribe. How the tribe chooses to store and use such data is at their discretion; most have an office of tribal enrollment where administrative staff have access to such records. Using a tribal enrollment record was the preferred method for identifying a list for participation selection at each tribe, since it reflected the population of interest (i.e., enrolled tribal members). For TUPP, the tribal enrollment records were restricted to adults residing within the county or counties in which the reservation was located. However, not all tribes were able or interested in providing an enrollment report. If this was the case, the approach described below was utilized.Indian Health Service (IHS) User Population—IHS provides medical and public health services to members of federally recognized American Indian tribes. Registered American Indian patients that had at least one direct or contract inpatient stay or outpatient visit or a direct dental visit (as recorded in the IHS central database) during the prior three years within a particular contract health service delivery area (CHSDA) were defined as users [21]. Thus, any American Indian community member who did not interact with any of the tribe’s health, dental, pharmacy, or human services during this timeframe were not included in the user population, whereas the utilization of the tribal enrollment records for sampling missed all other American Indian people living in the tribal nation. Both sampling methods were restricted to adults residing within the reservation boundaries.

To protect tribal members, TTUP staff were restricted from accessing sampling lists. TTUP staff guided tribal partners on restricting the lists based on additional eligibility criteria (i.e., 18 years of age and above, lives in the partner service area) and then tribal staff worked with IHS staff to generate a random sample list based on patient services.

#### 4.2.3. Sampling Methodology for Urban Communities

Tribal enrollment lists and IHS user population data were not available for sampling due to the diversity of the urban American Indian community, which reflects American Indian persons from hundreds of different American Indian tribal nations and no comprehensive medical or human services provider. Instead, respondent-driven sampling (RDS) [18] methods were adapted. RDS was the initial sampling protocol of choice for the urban population due to previous use in the TTUP I study [8] and proven success in hard to access or “hidden” populations. In brief, RDS combines “snowball sampling”, whereby individuals recruited to participate (i.e., seeds) refer a specific and limited number of participants with a mathematical model that weights the sample to reduce biases associated with referral method sampling. Based on recommendations for RDS [22], approximately 5 “seeds’’ were used as primary participants. Seeds were selected based on geographic location, gender, age, socioeconomic status, and smoking status. Each seed received three coupons to be handed out to individuals within their networks; from there, participating individuals then handed out more coupons. A USD 5 gift card was given to participants for every individual they recruited to participate, up to a total of USD 15.

As a result of COVID-19, the urban sampling and interviews became virtual. The community events and in person recruitment that had supported the use of RDS in TTUP I were unavailable, and there was a low recruitment of survey participants. After receiving community feedback that RDS was not working in the context of increased isolation due to COVID-19, the decision was made to move to a quota sample. Any eligible community member could participate in this, up to a certain threshold in demographic categories, based on proportional gender, age, and geographic location to the American Indian community in the urban area. Participants still received incentives for participating, and recruitment was still incentivized with USD 5 gift cards given for each individual recruited, up to USD 15.

#### 4.2.4. Adaptability to COVID-19 Constraints

Due to the COVID-19 pandemic that coincided with TTUP II implementation, all in-person interviews and activities were initially suspended while TTUP staff worked to provide a new virtual format for data collection. Much thought and consideration were given due to the changing of the modality of data collection, such as integrity of data, overall funding, and completion timeline, while honoring and respecting the wishes of participating tribal partners. The following are the protocol changes that were made to original protocols:Participant interviews were scheduled and completed by phone and/or video conference by trained interviewers;TTUP staff met and conducted all partnership correspondence (tribal leadership meetings, tribal site supervisor meetings, community interviewer meetings, and COG meetings) via phone and/or video conference;Trainings were conducted via video conferencing; study supplies and materials were emailed and/or mailed ahead of time;Community interviewers’ confidentiality agreements were distributed and returned via email prior to community data collection;During all phone or video conference interviews, the interviewer read a reminder about confidentiality and the interviewer asked the participant to respond with A, B, C, D, etc., rather than stating a response;Urban sampling methodology shifted to a quota sample, as described above.

### 4.3. Dissemination Phase (January 2021 to May 2022)

The final phase of the TTUP II study was the dissemination period. Dissemination occurred concurrently throughout implementation as participating tribal nations finished data collection at various times. TTUP staff worked closely with the Community Oversight Group (COG) to create a dissemination plan for study findings. The priority during the dissemination phase was to share the tribal specific data and tribal specific reports with the participating tribal nations.

Upon the completion of data collection, TTUP staff developed community specific reports. These reports were shared within six months after the completion of community data collection. Key findings were presented either in-person or virtually to tribal health directors, tribal leadership and any other community stakeholders. This included comparisons to data collected earlier if the tribal partner had participated in TTUP I, allowing for the contextualization of results. At this point, any additional data needs were identified by the tribal nation. The tribe then was able to use the information at their discretion (e.g., share with the community, use data for future grant writing endeavors for additional health disparity funding).

Lastly, a statewide aggregate report is being created for dissemination. There will be important considerations for the interpretation of this report due to the relatively long interval of data collection (i.e., 2018 to 2021) and the change in data collection methods as well as potential changes in tobacco behavior due to the pandemic. The statistical analysis plan for the statewide report will consider sensitivity analyses that either compare estimates by year of data collection or adjust for year. The statewide aggregate report, and its dissemination at conferences and in papers, will include only the participating tribal nations who agreed to be included, and will not identify participating tribes or provide tribal-specific information.

## 5. Discussion

Surveillance, which has been defined as the “systematic collection, analysis, and interpretation of data, closely integrated with the timely dissemination of these data” [23], is a critical part of public health efforts seeking to reduce commercial tobacco use and its associated cancer and health burdens. Traditional public health surveillance methods may not be appropriate or useful for American Indian communities and populations for a number of reasons, including a lack of cultural knowledge and sensitivity of government or non-American Indian organizations and staff; methods, such as telephone interviewing that do not take into account lifestyles of some American Indian people; or use of sampling methods that do not yield sufficient participants to represent American Indian communities. The result has often been insufficient and/or misleading data for American Indian community health planning and intervention. Herein, we highlighted the multiple strengths of TTUP, which is a survey designed by American Indians for assessing tobacco use among American Indians, including (1) respecting and honoring tribal sovereignty; (2) providing tribal specific data; (3) addressing the two types of tobacco (e.g., commercial, traditional) and intentionally collecting and reporting data on both; and (4) responding to the needs of the tribes and communities involved.

We also provided an in-depth overview of the TTUP process by discussing the efforts and activities that went into each of the TTUP phases (e.g., planning, implementation, and dissemination). Importantly, all of this work was accomplished through a CBPR/RBR framework that involved tribes and urban American Indian communities as active partners and decision makers throughout all phases of TTUP. As a result of this framework, we were able to easily pivot to respond to each of our partner’s needs during the COVID-19 pandemic, which coincided with the implementation of TTUP. While TTUP was designed and executed with tribes and urban American Indian communities in a single state (i.e., Minnesota) and has not undergone a formal evaluation in regard to the techniques and/or approaches that were most critical for its success, we foresee that the description herein of TTUP will be helpful to other tribes and public health organizations serving American Indians across the U.S. who are interested in conducting culturally relevant tobacco surveillance.

Since TTUP was developed and implemented through partnerships with a number of tribal nations, it is likely worthwhile to discuss a few of our key strategies for building and sustaining trust. A critical strategy for building trust was our established relationships in the tribal nations. Specifically, we leveraged relationships with the existing tribal programs that provide resources and support for tribal nations to conduct commercial tobacco prevention and control efforts, while increasing access to and educating about the sacred and healing role of traditional tobacco. Another key strategy for building trust was to ensure that involvement in TTUP provided a clear and tangible benefit back to each tribal nation. The benefits include providing tribes with their own data as well as committing to assist tribes in using their data to pursue future grants or sustain programming needs to reduce commercial tobacco use and promote the sacred role of traditional tobacco. We also want to acknowledge that there is an underlying extensive network of connections, resources, and cultural norms and values, which is complex and challenging to appropriately detail here. For these reasons, TTUP has been depicted as a river. The trust and relationships built through TTUP represent the headwaters, the data collected from the multiple communities among hundreds of community members come together to represent a vibrant, flowing river, and our future work that will be informed by TTUP is represented by numerous tributaries.

## 6. Conclusions

There are well known cultural teachings across Indian Country, relating to decision making centering around the Seventh-Generation Principle as depicted in the Great Law of the Haudenosaunee, founding document and philosophy of the Iroquois Confederacy, after which the United States is partially modeled [24].

“*In our every deliberation, we must consider the impact of our decisions on the next seven generations…Look and listen for the welfare of the whole people and have always in view not only the present but also the coming generations, even those whose faces are yet beneath the surface of the ground the unborn of the future Nation*”.

This principle, while not ubiquitous among every tribe, is a very common teaching in Indian Country. It recognizes the inherent obligation of present generations to consider the exponential long-term consequences of every action; in this particular context, certain health behaviors, such as the utilization of commercial tobacco. What each of us does now has a ripple effect [25]. It is an individual’s responsibility to ensure we are laying a good foundation for the generations to come for opportunities to flourish and succeed. Throughout the entire TTUP II study, American Indian culture and relationality was at the forefront. Every step of the way, we never lost sight of the end goal for our future generations; for those that have not come about, we lay the groundwork for healthier generations and a better tomorrow. It is our hope that the information we have provided within this manuscript can help guide others of similar interests in endeavors to better the lives of all who are part of Indian Country.

## Data Availability

Not applicable.

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
