# Peer review of "Tribal Tobacco Use Project II: Planning, Implementation, and Dissemination Using Culturally Relevant Data Collection among American Indian Communities"

_ijerph, 2022, doi:10.3390/ijerph19137708_

Round 1
Reviewer 1 Report
The authors have made the indicated corrections to the paper.Author Response
Thank you for the additional review.
Reviewer 2 Report
General
Overall I am happy with the revisions made by the authors to the manuscript. However I still feel the manuscript requires further expansion in terms of the effectiveness of the techniques employed, what worked well, what didn’t work well, and lessons to be learned in terms of future research. The authors have responded by adding a useful paragraph that focuses on trust. However this is only one albeit important issue. As such the discussion requires further expansion. it is important to discuss the approach employed and implications in terms of undertaking similar research. I have made the following additional minor comments. I hope the authors find this useful in making a revision.
- Methods
Page 5 line 541
Tribal resolutions requires further explanation. Is it a traditional tribal written agreement that can be used to develop a formalised research agreement?
Page 6 line 628-629
I sincerely apologise if my comments appeared racist. This was not my intention. My experience with marginalised groups is that a larger proportion (but not all) would have low education levels, and as a result some may find it difficult to complete long surveys. This is an issue with the general population too, but it can be more significant in populations with lower education levels. The inclusion of the length of time is useful. I appreciate it isn’t an issue in your population.
The issue of language may be important if AI prefer to use their own language to communicate. Could the authors clarify this to provide justification for the approach of using only English.
Page 6 line 655
A short sentence in terms of the possible impact of the incentive on sample bias, and perhaps of respondent driven sampling may have controlled for this.
Discussion p8 line 360-385
As stated in my general comments, I do not feel the revisions address the issues I have raised. The inclusion of trust and trust building is a welcome and informative addition. However readers need to be informed in terms of other key issues to consider in conducting similar research. Were there any feedback mechanisms built into the process to seek feedback from stakeholders in terms of what if anything could have be done differently? In terms of the research findings are there any lessons that can be learnt in terms of how to reduce smoking prevalence and implementing smoking prevention initiatives.
Reviewer 3 Report
No further comment.Author Response
Thank you for the additional review.
Round 2
Reviewer 2 Report
Discussion line 431- this should be reworded. i do not see how this can serve as a blueprint when we do not have any information on how well the process worked. I do not feel my origional comment about the discusssion has been adequately addressed.
Author Response
Thank you for the additional comment. We have removed the word 'blueprint' from the discussion section and the title.
This manuscript is a resubmission of an earlier submission. The following is a list of the peer review reports and author responses from that submission.
Round 1
Reviewer 1 Report
The paper presented is intended to describe the planning, implementation, and dissemination of a project targeted to a specific population. Therefore, turns out to be a novel, well-written paper that fulfills the stated objective.
I only have one comment, since at the end it´s not clear what has been so far the project scope, and if it specifically has achieved the impact on the prevalence of commercial tobacco consumption, on the interest in quit smoking, and on the abstinence rates from the instrumentation of programs for quit smoking through the project described.
Reviewer 2 Report
Comments to authors: Tribal tobacco use project II: A blueprint for planning, implementation, and dissemination using culturally relevant data collection among American Indian communities
General
This paper outlines the methodology employed to tackle the issue of tobacco use among American Indian communities. This is a valuable piece of research and it has implications in terms of undertaking similar research among hard to engage populations. It is also interesting and informative as it describes the cultural importance of tobacco use among American Indian culture, and the importance of distinguishing between traditional and commercial tobacco use. Tobacco prevalence is very high among American the Indian population, and this approach is valuable in terms of developing culturally appropriate interventions.
The process that was employed to engage with the Indian community including key stakeholder groups is outlined in significant detail, in addition to the issues considered in developing data collection tools. However, there is very limited information in terms of the effectiveness of the techniques employed, what worked well, what didn’t work well, and lessons to be learned in terms of future research. Indeed the discussion is extremely short. Working in partnership with a number of stakeholder groups can be difficult and it would be useful to know what worked for example in terms of building trust, and also if there were any blocks or barriers that had to be overcome. The discussion should be significantly expanded to address these issues. For this reason I am suggesting that the authors submit a major revision, since it includes the rewriting of one key section of the paper. Overall the paper reads well and I did find it interesting. However to increase the value in terms of a wider audience, it is important to discuss the approach employed and implications in terms of undertaking similar research. I have made the following comments to particular sections of the text and I hope the authors find this useful in making a revision.
- Introduction
Page 2 line 48-89
Need information on the size of the American Indian population, including Minnesota. How do smoking prevalence rates for American Indians in Minnesota compare with other Indian populations? Are there any reasons why rates are higher in the northern plains? Include some background in terms of how they live their lives. E.g. do they live as a distinct community or do they assimilate with the general population etc.
Page 2 line 90-95
Is there any data on the prevalence of traditional tobacco use. Would it not be important to find out about this as well?
Page 3 line 98
Insert a short description of what the first project did.
Page 4 line 150-173
Again a clearer description between the current phase and the previous TTUP phase of the research is required.
- Methods
Page 4 line 174-179
This is a long timeframe which means attitudes, values, and prevalence may be different due to different time periods. This needs to be discussed in terms of how this may limit analysis of the data.
Page 4 line 181-183
This is a long planning period. This needs to be addressed in the discussion section, perhaps highlighting the need to invest a lot of time into the planning phase of similar research.
Page 5 line 215-217
Clarify whether these were members of the American Indian population.
Page 5 line 218
Need to describe better what a tribal resolution is. Is this a research agreement put within the framework of tribal culture?
Page 5 line 234-244
This does seem like a long survey. Would American Indian populations be different to the general population in terms of attention span etc that may have needed to be considered? What about language? Was it available in different languages and was this considered by the research?
Page 6 line 266
Need to expand the role of incentive in the discussion.
Page 6 line 284-288
IS this a representative database? The rationale for using this database needs further explanation. Surely the Tribal Enrolment Records would have been more representative.
Page 6 line 294-295
Need further explanation in terms of why lists were not available in urban areas.
Discussion p8 line 360-385
As stated in my general comments, this needs to be expanded. Readers need to be informed in terms of the key success factors and the key issues to consider in conducting similar research. Were there any feedback mechanisms built into the process to seek feedback from stakeholders in terms of what if anything could have be done differently? This needs to be discussed.
In terms of the research findings are there any lessons that can be learnt in terms of how to reduce smoking prevalence and implementing smoking prevention initiatives.
Reviewer 3 Report
This paper focuses on the approaches deployed in a survey to obtain tobacco use for the minority community. The title of the manuscript is rather misleading considering the objective of this presentation is on survey criteria and its approaches, and the focus is targeted to a community in one country. The limitation and disadvantages of the approach may need to be shared.